# FROM UNCERTAINTY TO INCONSISTENCY: OPEN-SET RF FINGERPRINT IDENTIFICATION

## ABSTRACT

The rejection of unknown devices outside the known categories is crucial for radio frequency fingerprint identification (RFFI). Current open-set recognition (OSR) methods rely on the uncertainty of the model output, where unknown classes exhibit low confidence and vice versa for known classes. However, we demonstrate that uncertainty-based methods face a significant challenge, particularly in RFFI, which is termed "Overconfidence on Unknown Signal Segments" (OUSS), where unknown signal segments are misclassified with high confidence, directly contradicting the expected low-confidence characteristic for unknown classes. Inspired by an interesting observation that predictions for unknown classes across multiple models exhibit high inconsistency, while known classes exhibit the opposite, we propose to leverage decision entropy and max-agreement consensus to quantify the inconsistency. Based on the decision entropy and the max-agreement consensus, we propose an inconsistency based open-set RFFI approach (IncOS-RFFI). We conduct extensive experiments on the seven open-source radio frequency fingerprint datasets with seventeen benchmarks and demonstrate the effectiveness of our proposed IncOS-RFFI compared to existing OSR algorithms.

## 1 INTRODUCTION

Radio frequency fingerprint identification (RFFI) is a powerful technique for authenticating and tracking wireless transmitters by leveraging subtle waveform distortions introduced by device-specific hardware imperfections Zhang et al. (2023); Adesina et al. (2022). Early RFFI approaches primarily relied on handcrafted signal features designed by experts, but such methods exhibit limited scalability, poor robustness to environmental changes and insufficient generalizability to unknown scenarios Sa et al. (2019); He & Wang (2020); Zhang & Li (2023). To overcome these limitations, deep learning technology has been introduced into RFFI to automatically learn discriminative features from radio frequency fingerprints for improved recognition capabilities Zhou et al. (2021); Wang et al. (2020); Huang et al. (2017). Therefore, deep learning based RFFI has been widely studied under the closed-set assumption Zhou et al. (2021); Wang et al. (2020); Huang et al. (2017). However, with the rapid development of wireless technology, unknown devices may cause models trained only on known categories to fail. This challenge highlights that RFFI needs to not only identify known devices, but also be able to detect unknown devices to ensure the stability and security of monitoring Naik et al. (2020); Park et al. (2014). The open-set recognition (OSR) methods leverage the uncertainty of the model output to solve this problem, where unknown classes show low confidence while known classes show the oppsite Bendale & Boult (2016); Chen et al. (2021). Unfortunately, as shown in Figure 1 , we observed that these uncertainty based methods in RFFI suffer from a significant "Overconfidence on Unknown Signal Segments" (OUSS) challenge. Specifically, the uncertainty of model output for unknown classes is lower than expected, with up to 64.00% of the signal segments from unknown devices having extremely high confidences (*i.e.,* the maximum softmax probabilities are $> 0.96$) and being misclassified. In contrast, this proportion is only 24.90% in the CIFAR dataset.

Uncertainty based open-set RFFI methods can be categorized as single-model and multi-model approaches, all of which identify signal segments with high uncertainty in the output distribution as unknown. Single-model methods include OpenMax, generative methods, and prototype learning based methods. OpenMax Wu et al. (2023) extends the traditional softmax layer by fitting a Weibull distribution using extreme value theory (EVT) to compute open-set probabilities. However, due to

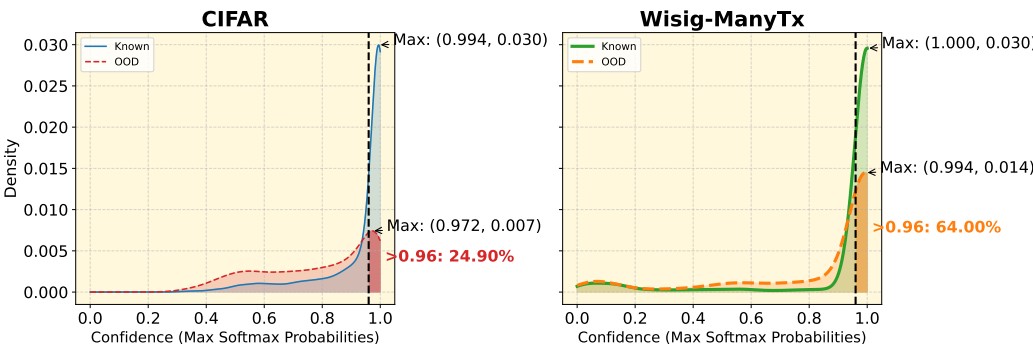

Figure 1: The Kernel Density Estimation of Confidence Distribution across Datasets. The CIFAR dataset is from Computer Vision and the WiSig-ManyTx dataset is from RFFI.

difficulties in tail modeling, it often fails to correct low uncertainty estimates for unknown signal segments, which leads to the OUSS challenge. Generative methods, such as generative adversarial networks (GAN) Guo et al. (2024) and variational autoencoders (VAEs) Karunaratne et al. (2021), detect unknown signal segments by generating anomalous signal segments. However, their limited representation of the unknown space fails to induce high uncertainty for unknown signal segments, while unstable training further distorts uncertainty calibration, leading to the OUSS challenge. Prototype learning based methods Wang et al. (2023a) calculate the distance between signal segments and class prototypes to perform open-set RFFI. However, ambiguous boundaries lead to low distance-based uncertainty for the OUSS signal segments. Compared with single-model methods, multi-model methods, such as Monte Carlo (MC) Dropout Justamante & McClure (2024) and deep ensembles Balasubramanian et al. (2021), detect unknown classes through the uncertainty of the average output distribution of multiple models. MC Dropout uses multiple random forward propagations, but the high correlation between propagations still leads to the OUSS challenge and the unknown signal segments are always misclassified as the same class. Deep ensembles leverage multiple independently trained models, however, for unknown and low-confidence known signal segments, the averaged output distribution may lead to low confidence, making it difficult to distinguish between unknown and low-confidence known signal segments.

We observe that predictions for unknown classes are often inconsistent across multiple independent models, while those for known classes tend to be consistent, even in the case of the OUSS challenge or when the confidences for known classes were low. This observation motivates us to propose an inconsistency based open-set RFFI method (IncOS-RFFI), as shown in Figure 3. In particular, IncOS-RFFI quantifies the prediction inconsistency among multiple models via two mechanisms, where decision entropy is utilized for comprehensive open-set detection and max-agreement consensus is leveraged for efficient inference. Notably, IncOS-RFFI is simple but efficient, and exhibits strong robustness to the OUSS signal segments.

The main contributions of this paper are summarized as follows.

- We first reveal and analyze the significant OUSS challenge in uncertainty based open-set RFFI methods.

- We propose an inconsistency based open-set RFFI method (IncOS-RFFI) in the presence of the OUSS challenge. IncOS-RFFI quantifies the prediction inconsistency across models by two mechanisms, where the decision entropy achieves comprehensive open-set detection (IncOS-RFFI-DE) and the max-agreement consensus achieves efficient inference (IncOS-RFFI-MA).

- We conduct extensive experiments using seven open-source radio frequency fingerprinting datasets to validate the effectiveness of the proposed approaches for open-set RFFI tasks. Experimental results demonstrate that our approaches outperform existing OSR methods in open-set detection metrics.

## 2 RELATED WORK

### 2.1 OPEN-SET RFFI BASED ON UNCERTAINTY

OSR was first formalized in Scheirer et al. (2012). Later OpenMax is proposed by combining deep learning with OSR Bendale & Boult (2016). Existing open-set RFFI has made significant progress by exploiting the uncertainty of the model output distribution to discriminate unknown classes. These schemes can be categorized into single-model and multi-model schemes.

#### 2.1.1 SINGLE-MODEL OPEN-SET RFFI

Single-model open-set RFFI methods can be categorized into OpenMax-based, generative methods, and prototype learning based methods. Inspired by OpenMax, an open-set RFFI framework was proposed to combine feature distance, triplet loss, and EVT for open-set RFFI Wu et al. (2023). Furthermore, slice-based preprocessing and noise augmentation are incorporated to enhance the framework Zhang et al. (2022). Other studies, such as NS-RFF Xie et al. (2021) and HyperRSI Fu et al. (2024), utilize hypersphere representations to enhance the ability to distinguish known from unknown devices. Furthermore, generative methods Karunaratne et al. (2021); Wang et al. (2023b) aim to enhance training by generating unknown data. Prototype learning based methods Wang et al. (2023a) reject unknown devices through feature-prototype similarity and EVT modeling. However, most uncertainty based single-model methods are prone to misclassification errors due to the OUSS challenge. OpenMax-based confidence adjustment is less effective for the OUSS signal segments, and generative models may fail to generate signals that truly represent the unknown space. Furthermore, prototype learning based models are prone to ambiguous rejections on the OUSS signal segments near the decision boundary. In contrast, our IncOS-RFFI approach based on inconsistency leverages the decision entropy and the max-agreement consensus of multiple models to detect unknown classes, avoiding reliance on distribution fitting, generative capability, or prototype distance. This design enhances robustness to the OUSS challenge while minimizing inference overhead.

#### 2.1.2 MULTI-MODEL OPEN-SET RFFI

Uncertainty based multi-model approaches are less explored and typically rely on averaging the output distributions of multiple models. MC Dropout Justamante & McClure (2024) estimates the uncertainty of the averaged output distribution through multiple random forward propagations. However, models are often highly correlated during random forward propagation, leading to the OUSS challenge, where each model consistently misclassifies signal segments into the same class. In contrast, our IncOS-RFFI method leverages the inconsistency to effectively address this problem.

### 2.2 OSR IN COMPUTER VISION

In computer vision (CV), OSR primarily encompasses discriminative models, generative models, and prototype based methods. Discriminative approaches, such as DOC Shu et al. (2017) and OLTR Liu et al. (2019), primarily focus on boundary modeling for open-set detection. Generative approaches, such as OpenMatch Saito et al. (2021), OSRCI Neal et al. (2018), and CROSR Yoshihashi et al. (2019), combine reconstruction or counterfactual generation to represent unknown spaces. Prototype-based approaches, such as GCPL Yang et al. (2018), RPL Chen et al. (2020), ARPL Chen et al. (2021), MPF, AMPF, and AMPF++ Xia et al. (2023), achieve effective OSR through feature-prototype similarity and adversarial mechanisms. While these approaches perform well in image classification tasks, they often perform poorly in RFFI due to the OUSS challenge, and suffer high computational cost and unstable training. We also employ several CV based OSR methods as baselines for comprehensive evaluation.

## 3 PROBLEM DEFINITION

Open-set RFFI can be formulated as a $K + 1$ classification problem to correctly classify signal segments from $K$ known classes while detecting signal segments from unknown classes. Let the training dataset be denoted as $\mathcal{D}_{train} = \{(\boldsymbol{x}_i, y_i) \mid i \in 1, \ldots, L, y_i \in 1, \ldots, K\}$, where $L$ is the total number of labeled signal segments from known classes. The testing dataset is represented as

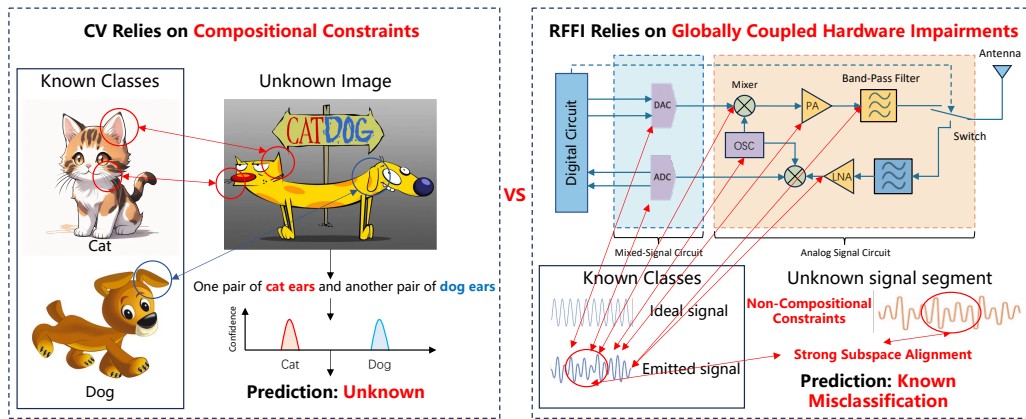

Figure 2: The Data Characteristics of Radio Frequency Fingerprints Compared to Images.

$\mathcal{D}_{test} = \{\boldsymbol{x}_j \mid j \in 1, \ldots, U\}$, which includes both signal segments from known classes $\{1, \ldots, K\}$ and signal segments from unknown classes. Here, $U$ denotes the total number of test signal segments. Each $\boldsymbol{x}_i$ refers to the $i$-th signal segment in $\mathcal{D}_{train}$ with the corresponding label $y_i$, while $\boldsymbol{x}_j$ is the $j$-th signal segment from $\mathcal{D}_{test}$. Both $\boldsymbol{x}_i$ and $\boldsymbol{x}_j$ are extracted from the received signal through sampling. A typical signal segment $\boldsymbol{x}_i$ can be written as

$$\boldsymbol{x}_i = \begin{bmatrix} r_I[1] & r_I[2] & \ldots & r_I[M] \\ r_Q[1] & r_Q[2] & \ldots & r_Q[M] \end{bmatrix}, \tag{1}$$

where $M$ denotes the number of sampled points within a segment, and $r_I[\cdot]/r_Q[\cdot]$ represent the in-phase/quadrature components sampled from the received signal $r(t)$. It is defined as

$$r(t) = h\phi(s(t)cos(\omega_0 t + \theta)) + n(t), \tag{2}$$

where $s(t)$ is the baseband signal, $cos(\omega_0 t + \theta)$ is the carrier modulated by center frequency $\omega_0$ and phase $\theta$, the parameter $h$ characterizes the wireless channel, $n(t)$ is additive white Gaussian noise (AWGN) with zero mean and variance $\sigma_n^2$, The function $\phi(\cdot)$ models the hardware-introduced signal variations and captures the unique radio frequency fingerprint of the emitter.

## 4 METHODOLOGY

In this section, we explore the cause of the aforementioned OUSS challenge and introduce an inconsistency based open-set RFFI approach (IncOS-RFFI), where IncOS-RFFI quantifies the prediction inconsistency among models via decision entropy and max-agreement consensus. For ease of understanding, we start with the exploration of the cause of the OUSS challenge, and then present an overview of the proposed IncOS-RFFI, followed by the details of these approaches for OSR-RFFI tasks.

### 4.1 CAUSE OF THE OUSS CHALLENGE

Visual classification often relies on compositionality, where an image $\mathbf{I}$ of class $y$, whose feature representation is composed of multiple semantic parts (*e.g.,* head, ear, tail) Lee et al. (2019). Let $\mathcal{P}_i$ denote the $i$-th semantic part, and $Q$ be the total number of parts. The overall feature representation $\mathcal{F}(\mathbf{I})$ can be written as

$$\mathcal{F}(\mathbf{I}) = \bigoplus_{i=1}^{Q} \varphi_i(\mathcal{P}_i) \otimes \Psi(\{\mathcal{P}_i\}_{i=1}^{Q}), \tag{3}$$

where $\varphi_i(\mathcal{P}_i)$ extracts the local feature of the $i$-th semantic part, $\bigoplus$ denotes the fusion operation, and the operator $\otimes$ denotes a gating operation that modulates the part features according to the compositional constraints $\Psi$, which evaluates the validity of the composition as

$$\Psi(\{\mathcal{P}_i\}) = \begin{cases} 1 & \text{if the parts form a valid composition} \\ 0 & \text{otherwise (\emph{e.g.,} cat head + dog body)} \end{cases}. \tag{4}$$

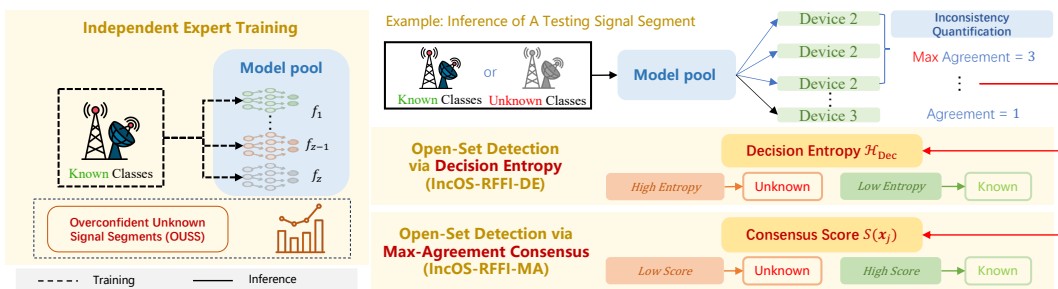

Figure 3: Overview of our IncOS-RFFI.

When $\Psi = 0$, the composite feature is assigned a low confidence due to the invalid part combinations.

The significant OUSS phenomenon in RFFI stems from the non-compositional entanglement characteristics of the radio frequency fingerprints. As shown in Figure 2, unlike CV where objects exhibit compositional strcuture (*e.g.,* a cat typically consists of eyes, ears, and a tail), radio frequency fingerprints arise from globally distributed and entangled hardware impairments without compositional constraints. The hardware impairment components of radio frequency fingerprints are not completely orthogonal and may be entangled. Therefore, while the dominant impairment portion is incomplete, it is sufficient for accurate closed-set classification. If the impairments of an unknown signal segment are highly similar to the dominant impairment portion of a known class but also contain other impairment components, the lack of componential constraints may cause the classifier to output high confidence in the the known class, which can easily lead to the OUSS challenge.

### 4.2 FRAMEWORK OVERVIEW

As shown in Figure 3, the proposed IncOS-RFFI framework consists of the independent expert training, the open-set detection via decision entropy, and the open-set detection via max-agreement consensus. For the independent expert training, a pool of $Z$ deep classifiers with random initialization, independent parameters, and independent data streams are trained on $\mathcal{D}_{train}$ to maximize predictive diversity. For the open-set detection, decision entropy is used to quantify the inconsistency among predictions from multiple models, where high entropy indicates unknown classes. Note that for faster inference speed and lower computational overhead, we introduce the max-agreement consensus to quantify the inconsistency, called IncOS-RFFI-MA.

### 4.3 INDEPENDENT EXPERT TRAINING

To improve the prediction diversity, a pool of $Z$ deep classifiers $\{f_z\}_{z=1}^{Z}$ with isolated dataflow and identical architecture are independently trained on $\mathcal{D}_{train}$. Specifically, the independent training means that each model is initialized with distinct random seeds, and no parameter sharing or gradient exchange occurs. Besides, the dataflow isolation means that each model has unique dataflow sequence. Furthermore, identical architecture ensures that the diversity among models, so that models develop unique decision boundaries for the OUSS challenge.

### 4.4 OPEN-SET DETECTION VIA DECISION ENTROPY

For the $j$-th input signal segment $\boldsymbol{x}_j \in \mathcal{D}_{test}$, we denote the softmax output distribution of the $z$-th model as $\boldsymbol{p}_z^{(j)} = f_z(\boldsymbol{x}_j)$ and denote the prediction as $y_z^{(j)} = \arg\max(\boldsymbol{p}_z^{(j)})$. Each expert's category prediction provides a direct indicator of whether the models agree or disagree on the same class. We define the decision frequency as a vector $\boldsymbol{v} = [v_1, \ldots, v_K]$, where each component $v_k = \frac{1}{Z} \sum_{j=1}^{Z} \mathbb{I}(y_z^{(j)} = k)$ represents the prediction frequency for class $k$. And $\mathbb{I}(\cdot)$ is the indicator function. To quantify the inconsistency among predictions, we define decision entropy based

on the decision frequency as

$$\mathcal{H}_{\text{dec}} = -\sum_{k=1}^{K} \boldsymbol{v}_k \log \boldsymbol{v}_k, \tag{5}$$

where low entropy indicates known signal segments with high consistency, high entropy indicates open-set signal segments with high inconsistency. Hence, the corresponding open-set detection rule can be expressed as

$$\text{Decision}(\boldsymbol{x}_j) = \begin{cases} \text{Known Class } \hat{y}^{(j)}, & \mathcal{H}_{\text{dec}} < \tau_H \\ \text{Open-Set}, & \mathcal{H}_{\text{dec}} \geq \tau_H \end{cases}, \tag{6}$$

where

$$\hat{y}^{(j)} = \arg\max_k \sum_{z=1}^{Z} \mathbb{I}(y_z^{(j)} = k), \tag{7}$$

and $\tau_H$ denotes the entropy threshold.

### 4.5 Max-Agreement Consensus

While the aforementioned decision entropy provides an effective criterion for open-set RFFI, its computational cost can be prohibitive for resource-constrained deployment scenarios. Specifically, the consensus score is defined as $S(x_j) = \max_{k \in 1,...,K} \sum_{z=1}^{Z} \mathbb{I}(y_z^{(j)} = k)$, which is a substantially simpler metric and can serve as an efficient surrogate while delivering comparable performance. These two metrics, $\mathcal{H}_{\text{dec}}$ and $S(x_j)$, exhibit a strong inverse correlation; a high $S(x_j)$ (signifying high consensus) corresponds to a low $\mathcal{H}_{\text{dec}}$, and vice versa. This relationship is underpinned by the theoretical lower bound of $\mathcal{H}_{\text{dec}}$ for a given $S(x_j)$, which can be expresses as

$$\mathcal{H}_{\text{dec}} \geq -\frac{S(x_j)}{Z} \log \frac{S(x_j)}{Z}$$
$$-\frac{Z - S(x_j)}{Z} \log \frac{Z - S(x_j)}{Z}. \tag{8}$$

Consequently, under typical conditions where multiple models clearly distinguish between known and open-set signal segments, a decision boundary for $S(x_j)$ can closely approximate for $\mathcal{H}_{\text{dec}}$.

The primary advantage of this substitution is a significant gain in computational efficiency. Calculating $S(x_j)$ only involves integer counting and a max operation, with a time complexity of $\mathcal{O}(C)$. In contrast, computing $\mathcal{H}_{\text{dec}}$ requires $C$ logarithmic operations and floating-point multiplications, which are computationally expensive on hardware such as edge devices or embedded systems.

Therefore, for applications where inference speed and power consumption are critical, we recommend using $S(x_j)$ as a simplified and highly efficient alternative to $\mathcal{H}_{\text{dec}}$. Therefore, the open-set detection rule can be expressed as

$$\text{Decision-Base}(x_j) = \begin{cases} \text{Known Class } \hat{y}^{(j)}, & S(x_j) \geq \tau_S \\ \text{Open Set}, & S(x_j) < \tau_S \end{cases}, \tag{9}$$

where $\hat{y}^{(j)}$ is computed using (7), $\tau_S \in \mathbb{Z} \cap (1, Z)$ is a predefined consistency threshold.

### 4.6 Discussion: Why Inconsistency Helps Alleviate the OUSS Challenge

#### 4.6.1 Local Optima Lead to Predictive Diversity.

The standard training paradigm for the RFFI classifier is to optimize the point estimate solution $\boldsymbol{w}^*$, and the objective function can be expressed as

$$\min_{\boldsymbol{w}} \ \mathbb{E}_{(\boldsymbol{x}_i, y_i) \in \mathcal{D}_{train}}[\mathcal{L}(f(\boldsymbol{x}_i; \boldsymbol{w}), y_i)],$$

where $f(\cdot; \boldsymbol{w})$ is the RFFI classifier mapping parameterized by $\boldsymbol{w}$, and $\mathcal{L}(\cdot, \cdot)$ is the cross-entropy loss function. During the non-convex optimization process for $Z$ models, the optimization ultimately converges to $Z$ local minima $\{\boldsymbol{w}_z^*\}_{z=1}^{Z}$, resulting in each model producing a different feature representation. For $\mathcal{D}_{train}$, despite good training performance, different models exhibit different feature

representations. Besides, the decision boundary of the classifier gradually solidifies during the optimization process, meaning that the model tends to classify the input as one of the known categories. However, the unknown signal segments may overlap with different known classes in the feature space of multiple models, so that different models classify unknown signal segments into different known categories, which leads to the diversity of model predictions.

### 4.6.2 Prediction Inconsistency in Unknown Classes.

As illustrated in Figure 4, the predictions of the 100 models for unknown signal segments exhibit notable inconsistency, suggesting the predictive diversity of unknown classes. In contrast, for known signal segments, except for signal segments 0, 28, and 29, the predictions of the remaining ones on the 100 models are consistent, which may indicate that the training process has enabled them to capture stable feature representations and decision boundaries for known categories. These observations imply that predictive inconsistency could serve as a potential metric for distinguishing between known and unknown classes, where lower consistency might be associated with signal segments from unknown categories.

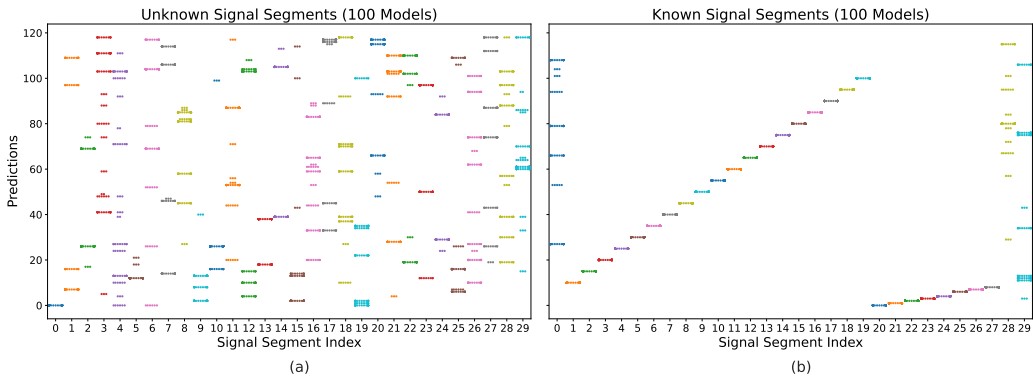

Figure 4: A Swarm Plot of 100 Model-Predictions For Randomly Selected 30 Known Signal Segments and 30 Unknown Signal Segments.

### 4.6.3 Advantages Over Deep Ensembles.

As shown in Figure 5, in deep ensembles (10 models), known signal segments with low confidence (*i.e.,* maximum softmax probability less than 0.8) account for 12.8% of all known classes. These low-confidence known signal segments are more easily confused with unknown classes. Figure 6(a) shows that the confidence scores of unknown signal segments are primarily distributed between 0.2 and 0.6, with values spread across the entire range from 0 to 1, while known signal segments with low confidence are concentrated in the 0–0.2 interval. In contrast, as shown in Figure 6(b), the inconsistency distribution exhibits an opposite trend: the maximum agreement count among the 10 models for unknown segments mostly ranges between 2 and 4, whereas that of low-confidence known segments lies between 7 and 10, indicating significantly lower inconsistency in known classes compared to unknown classes. This indicates that compared to deep ensembles, inconsistency based methods can effectively alleviate the misclassification problem caused by easily confused signal segments in deep ensembles.

## 5 Experiments

### 5.1 Dataset

We evaluate the performance of our approaches for open-set RFFI tasks using various public RFFI datasets shown in Table 1 of Appendix. As presented in Table 2 of Appendix, for each dataset, we select a group of devices as known classes $\mathcal{Y}_{\mathcal{K}}$ and set the other devices as unknown classes $\mathcal{Y}_{\mathcal{U}}$.

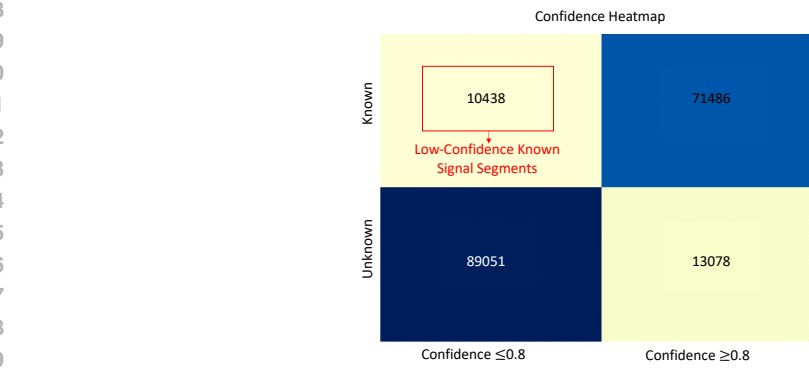

Figure 5: Confidence Heatmap of Deep Ensembles (10 models) on WiSig-ManyTx.

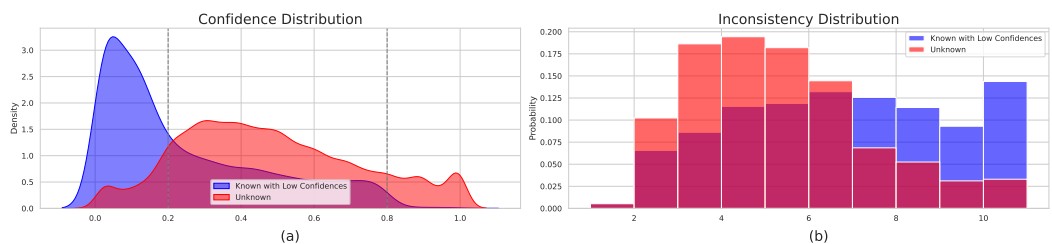

Figure 6: Comparative Analysis of Confidence and Inconsistency Distributions on 10 Models Between Unknown and Low-Confidence Known Signal Segments on WiSig-ManyTx.

## 5.2 EVALUATION METRICS AND IMPLEMENTATION DETAILS

Referring to the evaluation metrics in Chen et al. (2021); Lee et al. (2017); Dhamija et al. (2018), the closed-set classification rate (CCR) for known class accuracy, the area under the receiver operating characteristic (AUROC) and the open set classification rate (OSCR) for unknown detection capability, the area under the precision-recall curve (AUPR) class-imbalance robustness, and the detection accuracy (DTACC) for overall decision reliability are adopted. Besides, we leverage the ResNeXt-50 Xie et al. (2017) network as the backbone.

## 5.3 BASELINES

We compare the proposed IncOS-RFFI and IncOS-RFFI-Base with seventeen baselines, which can be grouped into five categories:

- Typical methods: the OpenMax Bendale & Boult (2016) based scheme, the DOC Shu et al. (2017) based scheme and the OLTR Liu et al. (2019) based schemes.

- Prototype learning based methods: GCPL Yang et al. (2018), RPL Chen et al. (2020), ARPL, ARPL with confusing samples (ARPL+CS) Chen et al. (2021), MPF, AMPF and AMPF++ Xia et al. (2023) based schemes.

- Generative methods: the CROSR Yoshihashi et al. (2019), the OSRCI Neal et al. (2018) and the OpenMatch Saito et al. (2021) based schemes.

- RFFI-specific methods: the NS-RFF Xie et al. (2021) and the HyperRSI Fu et al. (2024) based schemes.

- Multi-model methods: the MC dropout scheme Justamante & McClure (2024) and the deep ensembles.

| Methods | ManyTx | | SingleDay | | ORACLE | | FIT/CorteXlab | | POWDER | | Augmentation | | BlueTooth | |
|---|---|---|---|---|---|---|---|---|---|---|---|---|---|---|
| | CCR | AUROC | CCR | AUROC | CCR | AUROC | CCR | AUROC | CCR | AUROC | CCR | AUROC | CCR | AUROC |
| OpenMax | 88.94 | 83.63 | 99.65 | 92.05 | 99.20 | 90.84 | 83.82 | 50.68 | 99.99 | 52.02 | 83.20 | 50.98 | 51.83 | 46.94 |
| DOC | 88.60 | 82.91 | 87.60 | 85.24 | 99.99 | 88.41 | 85.58 | 51.88 | 99.99 | 50.62 | 83.20 | 17.02 | 52.35 | 47.87 |
| OLTR | 89.80 | 84.07 | 99.80 | 92.62 | 99.12 | 89.32 | 85.20 | 51.14 | 99.99 | 38.91 | 78.40 | 47.80 | 51.60 | 44.76 |
| GCPL | 89.57 | 84.47 | 99.80 | 89.45 | 99.99 | 81.57 | 84.90 | 46.23 | 99.99 | 83.19 | 38.00 | 46.09 | 51.79 | 41.33 |
| RPL | 89.85 | 86.23 | 99.66 | 90.47 | 99.99 | 84.81 | 85.59 | 50.42 | 99.99 | 29.33 | 36.00 | 49.43 | 47.15 | 46.57 |
| ARPL | 90.13 | 85.43 | 99.82 | 95.26 | 98.47 | 76.04 | 85.01 | 48.28 | 99.99 | 69.77 | 37.40 | 45.30 | 51.77 | 45.16 |
| ARPL+CS | 89.12 | 73.85 | 99.43 | 95.48 | 99.54 | 89.96 | 68.39 | 50.20 | 99.99 | 41.06 | 24.80 | 81.06 | 6.52 | 64.18 |
| MPF | 89.04 | 74.86 | 99.47 | 95.51 | 99.48 | 89.49 | 63.81 | 50.13 | 99.50 | 52.36 | 31.60 | 86.61 | 6.68 | 65.73 |
| AMPF | 89.31 | 78.60 | 99.54 | 96.17 | 99.48 | 91.02 | 66.22 | 48.96 | 99.99 | 50.84 | 32.20 | 85.35 | 6.68 | 63.96 |
| AMPF++ | 89.66 | 79.76 | 99.61 | 96.12 | 99.60 | 91.80 | 70.70 | 51.57 | 99.99 | 71.42 | 27.00 | 81.47 | 7.19 | 64.34 |
| CROSR | 87.46 | 72.45 | 98.35 | 85.02 | 98.79 | 79.21 | 82.75 | 46.01 | 99.99 | 48.50 | 81.96 | 45.12 | 50.21 | 46.75 |
| OSRCI | 83.14 | 80.17 | 97.29 | 89.42 | 97.52 | 87.53 | 58.98 | 51.63 | 99.99 | 26.19 | 43.80 | 47.45 | 8.10 | 58.75 |
| OpenMatch | 84.26 | 74.69 | 81.31 | 71.41 | 99.93 | 87.99 | 75.52 | 39.56 | 99.99 | 28.60 | 67.30 | 20.80 | 28.63 | 88.54 |
| NS-RFF | 85.08 | 69.20 | 98.83 | 78.55 | 99.66 | 89.13 | 76.56 | 50.44 | 96.00 | 65.76 | 38.60 | 51.95 | 51.33 | 43.53 |
| HyperRSI | 84.19 | 50.00 | 99.71 | 67.03 | 99.91 | 90.41 | 97.11 | 51.04 | 99.99 | 44.08 | 77.60 | 43.73 | 48.06 | 50.00 |
| MC Dropout | 89.75 | 90.13 | 99.72 | 95.50 | 99.42 | 82.74 | 86.97 | 52.31 | 50.00 | 53.21 | 84.20 | 46.78 | 51.12 | 39.08 |
| Ensemble | 90.84 | 90.50 | 99.81 | 96.31 | 99.99 | 88.87 | 86.34 | 52.43 | 99.99 | 63.19 | 84.20 | 46.85 | 53.04 | 39.37 |
| **IncOS-RFFI-MA** | 90.21 | 91.04 | 99.72 | 96.41 | 99.99 | 91.97 | 86.32 | 52.77 | 99.99 | 75.00 | 84.20 | 50.78 | 52.98 | 48.23 |
| **IncOS-RFFI-DE** | 90.33 | 91.12 | 99.75 | 96.52 | 99.99 | 92.01 | 86.34 | 52.82 | 99.99 | 75.12 | 84.20 | 50.82 | 53.01 | 48.35 |

Table 1: CCR and AUROC Comparison on seven radio frequency fingerprint datasets. The best method is emphasized in **bold**, and the underlined represents the second best result.

| Methods | CCR | AUROC | OSCR | DTACC | AUIN | AUOUT |
|---|---|---|---|---|---|---|
| MC Dropout Consistency | 26.50 | 50.00 | 13.25 | 50.00 | 72.26 | 77.74 |
| Multi-Head Consistency | 89.12 | 51.49 | 46.00 | 51.49 | 72.56 | 75.11 |
| **IncOS-RFFI-MA** | 90.21 | 91.04 | 86.31 | 90.20 | 91.68 | 89.51 |
| **IncOS-RFFI-DE** | **90.33** | **91.12** | **86.88** | **90.34** | **91.73** | **89.99** |

Table 2: Performance Comparison of MC Dropout Based Consistency, Multi-Head Based Consistency and Our InOS-RFFI on WiSig-ManyTx.

## 5.4 Overall Performance Comparison

As shown in Table 1, our proposed IncOS-RFFI consistently outperforms the open-set metrics. Although deep ensembles are highly competitive, they are slightly inferior to our approach in terms of AUROC while our approach achieves comparable CCR results to theirs, which shows that our inconsistency based method has alleviated the confusion problem in deep ensembles to some extent. Moreover, RFFI-specific schemes exhibit suboptimal performance on these datasets, primarily due to limited generalization of their signal-type tailored features. Furthermore, due to instability, generative methods are not as effective as typical methods and prototype learning based methods. Furthermore, the prototype learning based methods exhibit poor performance of CCR on the Augmentation and BlueTooth datasets, which shows that inherent characteristics of different datasets strongly affect the open-set discrimination capability of these methods. Although their AUROC is higher, the goal of OSR is to improve the open-set detection ability without compromising the classification accuracy of known classes, rather than sacrificing the classification accuracy of known classes.

## 6 Conclusion

In this paper, we have developed an InOS-RFFI approach to identify known devices, detect unknown devices and mitigate the OUSS challenge in RFFI. We independently train a pool of deep classifiers with independent parameters, isolated dataflow and identical architecture to promote predictive diversity. Moreover, to mitigate the confusion between unknown and low-confidence known classes, we propose that the signal segment with adequately high decision entropy or max-agreement consensus is indicated as known. Simulation results have demonstrated that our proposed InOS-RFFI-MA and InOS-RFFI-DE significantly outperforms existing benchmark algorithms in open-set RFFI tasks.

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

# A APPENDIX

In this section, we evaluate the effectiveness of IncOS-RFFI in open-set RFFI. We conduct extensive experiments on a large-scale radio frequency fingerprinting datasets and compare our method with several state-of-the-art open-set recognition approaches. The evaluation focuses on both closed-set classification accuracy and the ability to detect unknown emitters.

## A.1 DATASET

We evaluate the performance of our approaches for open-set RFFI tasks using various public RFFI datasets shown in Table 5. As presented in Table 3, for each dataset, we select a group of devices as known classes $\mathcal{Y}_{\mathcal{K}}$ and set the other devices as unknown classes $\mathcal{Y}_{\mathcal{U}}$.

In order to mitigate the impact of varying emitter-receiver distances on the radio frequency fingerprint, we select only the 20ft data from the ORACLE dataset for experimentation. To avoid the influence of different collection dates, we use only the data collected on January 8, 2019, from the FIT dataset. For the BlueTooth dataset, we focus exclusively on the mobile data collected at 250 Msps for our experiments.

## A.2 EVALUATION METRICS

Referring to the evaluation metrics in Chen et al. (2021); Lee et al. (2017); Dhamija et al. (2018), the closed-set classification rate (CCR) for known class accuracy, the area under the receiver operating characteristic (AUROC) and the open set classification rate (OSCR) for unknown detection capability, the area under the precision-recall curve (AUPR) for class-imbalance robustness, and the detection accuracy (DTACC) for overall decision reliability are adopted. Besides, we leverage the ResNeXt-50 Xie et al. (2017) network as the backbone.

- **CCR:** Accuracy on test signal segments from known classes under a specified threshold.

| Dataset | Classes | | Signal Segments | |
|---|---|---|---|---|
| | $|\mathcal{Y}_\mathcal{K}|$ | $|\mathcal{Y}_\mathcal{U}|$ | $|\mathcal{D}_{train}|$ | $|\mathcal{D}_{test}|$ |
| **ManyTx Hanna et al. (2022)** | 120 | 30 | 327.4k | 184.1k |
| **SingleDay Hanna et al. (2022)** | 20 | 8 | 128k | 96k |
| **ORACLE Sankhe et al. (2019)** | 10 | 6 | 50k | 46k |
| **FIT/CorteXlab Morin et al. (2019)** | 15 | 6 | 45k | 36.9k |
| **POWDER Reus-Muns et al. (2020)** | 2 | 2 | 1k | 1.4k |
| **Augmentation Soltani et al. (2020)** | 5 | 5 | 2.5k | 3.5k |
| **BlueTooth Uzundurukan et al. (2020)** | 25 | 8 | 27.6k | 12.2k |

Table 3: A Summary of the Selection of Training and Testing sets.

- **AUROC:** AUROC is a widely used primary evaluation metric for evaluating OSR performance by plotting the true positive rate against the false positive rate across varying thresholds Neal et al. (2018); Fawcett (2006); Davis & Goadrich (2006); Lee et al. (2017).

- **OSCR:** OSCR evaluates the trade-off between classification accuracy on known classes and the rejection rate of unknown classes, plotting CCR under various thresholds versus the false positive rate of unknowns Dhamija et al. (2018).

- **AUPR:** The precision-recall curve is graph plotting precision $= TP/(TP + FP)$ against recall $= TP/(TP + FN)$ by varying a threshold. The AUIN (or AUOUT) is the AUPR where the in- (or out-of-) distribution samples are specified as positive.

- **DTACC:** This metric corresponds to the maximum classification probability over all possible thresholds. We assume that both positive and negative examples have equal probability of appearing in the test set, *i.e.,* $P(x \in P_{in}) = P(x \in P_{out}) = 0.5$.

## A.3 IMPLEMENTATION DETAILS

The detailed parameter settings of our experiments are shown in Table 4. Each signal segment has the dimension of $2 \times 256$, where the two rows correspond to the sampled $I$ / $Q$ components and the 256 columns correspond to the number of sampling points.

| Parameters | Value |
|---|---|
| Signal segment dimension | $2 \times 256$ |
| Batch size | 256 |
| The number of epochs | 20 |
| Learning rate | 0.01 |
| Loss function | Cross-entropy |
| Optimizer | SGD |
| Momentum of optimizer | 0.9 |
| Weight decay of optimizer | 0.00005 |

Table 4: Experimental Parameter Settings

| Dataset | Waveform | Transmitters | Receiver(s) | Day(s) | Sampling Rate | Frequency | Bandwidth | Synthetic /Real-world |
|---|---|---|---|---|---|---|---|---|
| **ManyTx Hanna et al. (2022)** | WiFi | 150 WiFi | 28 USRP | 4 | 25Ms/s | 2462MHz | 20MHz | Real-world |
| **SingleDay Hanna et al. (2022)** | WiFi | 28 WiFi | 10 USRP | 1 | 25Ms/s | 2462MHz | 20MHz | Real-world |
| **ORACLE Sankhe et al. (2019)** | WiFi | 16 USRP X310 | 1 USRP B210 | - | 5Ms/s | 2.45GHz | 80MHz | Synthetic |
| **FIT/CorteXlab Morin et al. (2019)** | IEEE 802.15.4 | 21 USRP N2932 | 1 USRP N2932 | 5 | 5Ms/s | 433MHz | - | Real-world |
| **POWDER Reus-Muns et al. (2020)** | 4G, 5G, WiFi | 4 base stations | USRP B210 | 2 | 4G/5G:7.69Ms/s, WiFi: 5Ms/s | 2.685GHz | - | Synthetic |
| **Augmentation Soltani et al. (2020)** | WiFi | 10 transmitters | - | - | 5Ms/s | - | - | Synthetic |
| **BlueTooth Uzundurukan et al. (2020)** | BlueTooth | 250Ms/s: 33 Smartphones Tektronix TDS7404 | - | 250Ms/s, 5Gs/s, 10Gs/s, 20Gs/s | - | 15MHz-100MHz | Real-world |

Table 5: Dataset Summary on Our Validation Experiments

Beside, we compare the proposed IncOS-RFFI and IncOS-RFFI-Base with seventeen baselines, which can be grouped into five categories:

(1) Typical methods:

- OpenMax (Bendale & Boult (2016)): OpenMax extends the typical softmax function to provide a confidence score for open-set recognition. It operates by calculating the distance from a sample to the class prototypes of the training set. For an unknown sample, it assigns a low confidence score, signaling it as an outlier, rather than a member of any of the known classes. This is effective in open-set recognition problems, where the model must also identify unknown classes.

- DOC (Shu et al. (2017)): The Deep Open Classification (DOC) framework uses deep learning for open-set classification by learning a representation that can distinguish between known and unknown classes. DOC works by introducing an open-set classification objective that encourages the model to reject outliers (unknown classes) while classifying known classes accurately. It also uses a margin-based loss function for distinguishing between classes.

- OLTR (Liu et al. (2019)): Open Long-Tailed Recognition (OLTR) is designed for handling open-set problems with long-tailed data. It incorporates a method for dealing with both class imbalance and open-set recognition by using memory-based prototypes and rejection regions for unknown classes.

(2) Prototype learning based methods:

- GCPL (Yang et al. (2018)): Generalized Class Prototype Learning (GCPL) introduces the concept of prototype learning to open-set recognition. It uses prototypes of classes and classifies input samples based on their distance to the class prototypes, effectively distinguishing between known and unknown classes.

- RPL (Chen et al. (2020)): Relational Prototype Learning (RPL) extends prototype learning by considering relational structures between classes. It is more flexible than traditional methods because it learns to recognize not just individual class prototypes but also the relationships between them, allowing for better open-set classification.

- ARPL (Chen et al. (2021)): The Adaptive Relational Prototype Learning (ARPL) method adapts to the data by learning relational prototypes in an open-set scenario. It aims to improve upon RPL by dynamically adjusting the prototype space based on incoming data, effectively handling unknown classes.

- ARPL+CS (Chen et al. (2021)): This is an extension of ARPL, where Confusing Samples (CS) are incorporated into the training to enhance the model's robustness. These confusing samples are added to the training set to challenge the model, making it more capable of distinguishing between known and unknown classes.

- MPF (Xia et al. (2023)): Multi-Prototype Learning with Feature Expansion (MPF) utilizes multiple prototypes for each class to capture diverse class characteristics. It helps in open-set recognition by allowing each class to have several representative prototypes that can better handle variations within classes.

- AMPF (Xia et al. (2023)): Adaptive Multi-Prototype Learning with Feature Expansion (AMPF) is an improvement over MPF. It uses adaptive methods to refine prototypes during the training process, dynamically adjusting them to accommodate variations in the data.

- AMPF++ (Xia et al. (2023)): This is an enhanced version of AMPF, further improving the adaptability of prototypes by incorporating additional features or advanced learning strategies to improve the discrimination between known and unknown classes.

(3) Generative methods:

- CROSR (Yoshihashi et al. (2019)): Class-Conditional Generative Model for Open-Set Recognition (CROSR) uses a generative approach to model the distributions of known classes and the unknown class. It generates samples from known classes and uses these generated samples for recognition. For unknown classes, the model assigns a low likelihood, indicating it is not part of the known classes.

- OSRCI (Neal et al. (2018)): Open-Set Recognition via Conditional Inference (OSRCI) is a generative approach that learns a model capable of discriminating between known and unknown classes by generating the conditional distribution of each class and applying it to test samples.

- **OpenMatch (Saito et al. (2021)):** OpenMatch is another generative-based approach for open-set recognition that utilizes a generative model to generate representations of the known and unknown classes. It adjusts the decision boundary by learning from both the training samples and the generative model, improving open-set classification.

(4) RFFI-specific methods:

- **NS-RFF (Xie et al. (2021)):** The Noise Suppressed Radio Frequency Fingerprint (NS-RFF) method is specifically designed for Radio Frequency Fingerprint Identification (RFFI). It applies noise suppression techniques to enhance the performance of RFFI in open-set recognition scenarios, ensuring better discrimination between known and unknown RF signals.

- **HyperRSI (Fu et al. (2024)):** HyperRSI is a specialized approach for RFFI that leverages hyperparameter optimization to improve the model's robustness and adaptability to different RF signal conditions. It is tailored to work with the inherent challenges of RF fingerprinting, such as noise and interference.

(5) Multi-model methods:

- **MC Dropout (Justamante & McClure (2024)):** Monte Carlo (MC) Dropout is a regularization technique where dropout is applied during both training and inference. This allows the model to approximate uncertainty in its predictions, which can be useful for open-set recognition. By using MC dropout, the model estimates the uncertainty for each class and can reject samples with high uncertainty as unknown.

- **Deep Ensembles:** This method uses multiple models (often trained independently) and aggregates their predictions to improve generalization and robustness. In open-set recognition, deep ensembles help reduce overconfidence in the prediction of unknown samples by providing a more diverse set of predictions from multiple models.

The network architectures of these baseline were modified to accommodate the one-dimensional structure of the signal.

| Methods | CCR | AUROC | OSCR | DTACC | AUIN | AUOUT |
|---|---|---|---|---|---|---|
| OpenMax | 88.94 | 83.63 | 85.27 | 77.48 | 78.34 | 86.04 |
| DOC | 88.60 | 82.91 | 82.97 | 84.29 | 66.02 | 88.54 |
| OLTR | 89.80 | 84.07 | 82.77 | 80.23 | 83.12 | 79.81 |
| GCPL | 89.57 | 84.47 | 82.28 | 84.51 | 78.71 | 80.71 |
| RPL | 89.85 | 86.23 | 84.31 | 85.61 | 85.20 | 80.57 |
| ARPL | 90.13 | 85.43 | 83.48 | 85.60 | 80.89 | 81.29 |
| ARPL+CS | 89.12 | 73.85 | 72.89 | 70.46 | 70.78 | 69.91 |
| MPF | 89.04 | 74.86 | 73.38 | 73.29 | 69.45 | 71.76 |
| AMPF | 89.31 | 78.60 | 77.52 | 75.58 | 76.85 | 73.05 |
| AMPF++ | 89.66 | 79.76 | 78.70 | 76.62 | 78.55 | 73.94 |
| CROSR | 87.46 | 72.45 | 71.74 | 67.12 | 67.2 | 75.85 |
| OSRCI | 83.14 | 80.17 | 76.21 | 78.30 | 82.63 | 73.03 |
| OpenMatch | 84.26 | 74.69 | 71.41 | 72.14 | 73.70 | 69.53 |
| NS-RFF | 85.08 | 69.20 | 64.18 | 67.21 | 57.49 | 70.62 |
| HyperRSI | 84.19 | 50.00 | 75.26 | 50.00 | 72.26 | 77.74 |
| MC Dropout | 89.75 | 90.13 | 85.48 | 85.15 | 83.43 | 78.41 |
| Ensemble | **90.84** | 90.50 | 85.51 | 89.62 | 91.12 | 81.02 |
| **IncOS-RFFI-MA** | 90.21 | 91.04 | 86.31 | 90.20 | 91.68 | 89.51 |
| **IncOS-RFFI-DE** | 90.33 | **91.12** | **86.88** | **90.34** | **91.73** | **89.99** |

Table 6: Performance Comparison on WiSig-ManyTx.

## A.4 In-depth Analysis on OUSS Subset of ManyTx dataset

We analyzed the ManyTx dataset, which has the largest number of categories. As shown in Table 6, our method outperforms other methods in open-set metrics. To validate our method's performance on OUSS signal segments, we select unknown signal segments with a confidence score exceeding 0.96 as unknown classes for experiments. As shown in Table 7, the results show that our method outperforms other methods in all metrics. In particular, when compared to deep ensembles, our AUOUT improves by over 10%, demonstrating the advantage of inconsistency based methods in classifying known and unknown classes.

| Methods | AUROC | OSCR | DTACC | AUIN | AUOUT |
|---|---|---|---|---|---|
| OpenMax | 77.01 | 72.15 | 71.84 | 84.34 | 65.79 |
| DOC | 78.40 | 76.61 | 82.01 | 76.65 | 76.02 |
| OLTR | 80.31 | 79.51 | 76.38 | 89.12 | 56.87 |
| GCPL | 81.36 | 79.81 | 81.91 | 86.95 | 58.80 |
| RPL | 83.58 | 82.25 | 83.11 | 90.65 | 59.53 |
| ARPL | 81.67 | 80.21 | 79.00 | 87.55 | 59.59 |
| ARPL+CS | 65.47 | 59.28 | 61.07 | 77.56 | 53.02 |
| MPF | 66.62 | 65.60 | 66.24 | 78.29 | 44.70 |
| AMPF | 71.67 | 70.98 | 68.96 | 83.41 | 46.63 |
| AMPF++ | 73.45 | 72.82 | 70.63 | 84.71 | 48.22 |
| CROSR | 67.08 | 68.28 | 63.52 | 78.78 | 52.27 |
| OSRCI | 78.06 | 74.88 | 76.93 | 89.38 | 50.14 |
| OpenMatch | 85.48 | 80.87 | 85.70 | 94.32 | 58.06 |
| NS-RFF | 64.40 | 60.08 | 62.92 | 73.83 | 44.41 |
| HyperRSI | 50.00 | 73.62 | 50.00 | 83.52 | 50.91 |
| MC Dropout | 79.73 | 83.64 | 81.42 | 85.64 | 53.00 |
| Ensemble | 87.07 | 84.00 | 86.85 | 94.35 | 61.38 |
| **IncOS-RFFI-MA** | 87.03 | 84.79 | 87.15 | 94.46 | 71.5 |
| **IncOS-RFFI-DE** | **87.14** | **85.12** | **87.52** | **94.52** | **71.8** |

Table 7: Performance Comparison on 64.00% of OUSS from WiSig-ManyTx dataset.

## A.5 ABLATION EXPERIMENTS

### A.5.1 ABLATION ON CONSENSUS MECHANISMS.

To validate our proposed IncOS-RFFI, we conduct ablation experiments to evaluate the effectiveness of our approach using inconsistency based MC Dropout and inconsistency based 10 different classification head models with the same backbone. We observe that the predictions of the two methods are generally consistent, both for known and unknown classes, indicating that the predictive diversity brought by multiple independently trained models is very necessary, as shown in the Table 2.

### A.5.2 THE IMPACT OF THE NUMBER OF MODELS.

As shown in the Figure 7, we find that the performance of the inconsistency based method gradually improves as the number of models increases. Notably, even when using only 5 models, our method shows good results and performs better than the 5-model deep ensembles in distinguishing between known and unknown classes, as illustrated in Table 8.

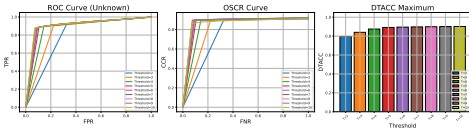

Figure 7: Comparison of ROC, OSCR Curves, and DTACC Plots Across Different Model Sizes on the WiSig-ManyTx.

| Methods | CCR | AUROC | OSCR | DTACC | AUIN | AUOUT |
|---|---|---|---|---|---|---|
| Ensemble | **90.67** | 89.51 | 87.06 | 89.57 | 91.07 | 80.97 |
| **IncOS-RFFI-MA** | 90.01 | 89.75 | 87.13 | 90.19 | 91.25 | 90.31 |
| **IncOS-RFFI-DE** | 90.15 | **90.03** | **87.20** | **90.31** | **91.51** | **90.43** |

Table 8: Comparison of Ours and the Deep Ensembles under five models

