# OpenReview forum: "From Uncertainty to Inconsistency: Open-Set RF Fingerprint Identification"
_ICLR.cc/2026/Conference — ICLR 2026 Conference Withdrawn Submission_

### Official Review · Reviewer_bxpf · 2025-10-25

**Soundness:** 2
**Presentation:** 3
**Contribution:** 2
**Rating:** 4
**Confidence:** 1

**Summary:**

The paper addresses open-set radio frequency fingerprint identification (RFFI), where models must both classify known emitters and reject unknown devices. The authors document a domain-specific failure mode for uncertainty-based OSR called Overconfidence on Unknown Signal Segments (OUSS): many unknown RF segments are misclassified with very high softmax confidence (e.g., >0.96), much more severely than in image datasets. They attribute this to non-compositional, globally entangled hardware impairments in RF signals (vs. part-based compositionality in vision), which can align strongly with a known class subspace even when the signal is unknown.

To mitigate OUSS, they propose IncOS-RFFI, an inconsistency-based detector built from a pool of independently trained classifiers. Two decision rules are presented: (i) decision-entropy over the ensemble’s argmax votes (IncOS-RFFI-DE) and (ii) a low-cost max-agreement threshold (IncOS-RFFI-MA). They show a lower bound linking entropy to agreement, justify that local minima diversity yields prediction variability on unknowns, and evaluate on seven RFFI datasets against 17 baselines (typical OSR, prototype-based, generative, RFFI-specific, and multi-model). IncOS-RFFI improves AUROC (unknown detection) while maintaining closed-set accuracy (CCR).

**Strengths:**

* Clear articulation of why standard OSR fails in RFFI (OUSS) via hardware impairment entanglement vs. compositional features in CV.
* Two intuitive signals (entropy and max agreement) with a lower-bound connection and rationale from local-optima diversity.
* Seven datasets, many baselines, including deep ensembles and MC-Dropout; consistent AUROC improvements without sacrificing CCR on most datasets.
* A low-compute proxy for entropy suitable for embedded/edge deployment (in principle).

**Weaknesses:**

1. Method effectiveness scales with Z models; paper lacks efficiency curves (AUROC/OSCR vs. Z), throughput/latency/energy numbers, and comparison to single-model OSR augmented with cheap post-hoc detectors (e.g., energy score, Mahalanobis) adapted to 1D I/Q.
2. No principled scheme for $\tau_H$/$\tau_S$ (e.g., target TPR/FPR on validation), and no calibration analysis despite OUSS being partly a calibration issue; could bias AUROC/OSCR across datasets.
3. The claim that independent training yields diversity is plausible but unquantified; no study of vote entropy vs. feature diversity or the effect of shared/backbone changes.
4. For deep ensembles, it’s unclear whether Z matches IncOS-RFFI’s Z, and how tuning parity was enforced across baselines (thresholding, early stopping, augmentations).

**Questions:**

1. How do AUROC/OSCR and CCR vary with Z = {2,4,8,16}? Please report latency, memory, and energy for DE vs. MA vs. deep ensembles on an edge-class device.
2. How are $\tau_H$ / $\tau_S$ chosen? Fixed operating point, or per-dataset validation? Could you report ROC/PR operating points and evaluate calibration (ECE/Brier) before/after your method?
3. Please provide pairwise disagreement / Q-statistic / vote entropy among experts and correlate with detection gains; include ablations where dataflow isolation is removed or architectures differ.
4. How does IncOS-RFFI compare to Energy-based OoD, Mahalanobis distance, or ODIN-style temperature scaling+noise (adapted to 1D I/Q) under matched compute?
5. Since some datasets are filtered by day/distance, can you test cross-receiver/day generalization (train/test splits across days/distances) to probe channel/receiver shift?

---

### Official Review · Reviewer_f6iu · 2025-10-30

**Soundness:** 2
**Presentation:** 2
**Contribution:** 2
**Rating:** 2
**Confidence:** 5

**Summary:**

The authors propose a method for radio frequency fingerprint identification that adopts an inconsistency-based open-set approach to address the issue of overconfidence on unknown signal segments, where such segments are often misclassified with high confidence. The method leverages decision entropy and max-agreement consensus to mitigate this problem.

**Strengths:**

1)	The Overconfidence on Unknown Signal Segments issue described in paper seems legitimate. Therefore, the motivation for this work is convincing.
2)	It seems feasible to use decision entropy and maximum consensus to solve the OUSS problem.
3)	The experimental results on various RFFI datasets are provided.

**Weaknesses:**

1)	Since the task in the paper is to detect the unknown frequency. How does the proposed method compare to the FFT-based approach?
2)	The proposed method is based on the decision entropy, which is a relative naive idea. Therefore, its novelty is somewhat limited.
3)	No references regarding the techniques used in the proposed method is provided in the method section.
4)	The authors should provide the proof for Eq (8).
5)	How to choose the entropy threshold values for different RFFI datasets to detect unknown frequnecies?
6)	Figure 7 is too small.

**Questions:**

The proposed method is only tested on the RFFI datasets. Is this method can also be applied to other time series data types?

---

### Official Review · Reviewer_G9KM · 2025-11-01

**Soundness:** 3
**Presentation:** 3
**Contribution:** 1
**Rating:** 4
**Confidence:** 3

**Summary:**

The paper addresses open-set recognition (OSR) for radio-frequency fingerprint identification (RFFI). The authors identify the “Overconfidence on Unknown Signal Segments” (OUSS) phenomenon, where deep RFFI models assign high confidence to unseen devices. To mitigate this, they propose IncOS-RFFI, an inconsistency-based method that quantifies disagreement among independently trained classifiers using two metrics: decision entropy (IncOS-RFFI-DE) and max-agreement consensus (IncOS-RFFI-MA). Experiments on seven open RFFI datasets show modest improvements over uncertainty-based OSR baselines and standard deep ensembles.

**Strengths:**

1.Clearly written and well-structured presentation.

2.Extensive benchmarking across multiple real-world RFFI datasets which merits mention.

3.The OUSS description provides a useful empirical characterization of overconfidence in signal-domain classifiers.

4.The approach is simple, reproducible, and practical for embedded or low-power scenarios.

**Weaknesses:**

1. Methodological novelty is limited: measuring cross-model inconsistency via entropy or vote agreement is a well-established ensemble-based OOD/OSR strategy (cf. Lakshminarayanan et al., 2017; Fort et al., 2021).

2. No new learning objective, architecture, or theoretical insight.

3. Reported gains over standard deep ensembles are small on most datasets (≤ 3 AUROC), with improvements confined to a few difficult domains.The experimental results demonstrate that IncOS-RFFI performs comparably to or slightly better than classical deep ensembles on most datasets, with meaningful gains on a few difficult RFFI benchmarks such as POWDER and Bluetooth. While this confirms that ensemble disagreement is a strong OSR signal in the RF domain, the improvements are incremental rather than transformative.

The one area where this approach might be adding robustness, might be noisy or cross domain problems — perhaps because disagreement patterns help detect unseen signal domains.

Ultimately, simply using the disagreement entropy  or  consesus metric to help alleviate high confidence on unknown samples is not sufficiently novel to be considered for an ICLR publication.
This is a solid applied study that documents overconfidence phenomena in RFFI and validates ensemble disagreement as an effective detection signal. However, the contribution is primarily empirical and domain-specific rather than methodological. The work would be more appropriate for a signal-processing or applied-ML venue than for ICLR.

**Questions:**

1. You state a theoretical lower bound for the decision entropy $H_{\text{dec}}​$ in terms of the max-agreement $S(x_j)$ (Eq. 8), but provide no derivation. Could you explicitly show how this bound is obtained and under what assumptions it holds (e.g., binary disagreement, fixed number of classes, or uniform dissenting distribution)? Also, does this bound remain tight when disagreement is distributed across multiple classes? Even straightforward applications of concave function properties should be at least mentioned.

2. You argue that decision entropy is computationally prohibitive for deployment on “resource-constrained” hardware. Could you provide quantitative runtime or power measurements that demonstrate this difference? For instance, how does computing $H{\text{dec}}$​ compare to $S(x_j)$ relative to the total inference time of the ensemble?

3. The link between model inconsistency and unknown-class detection has been extensively studied in ensemble and Bayesian uncertainty literature (e.g., Deep Ensembles, MCDropout). Could you clarify what is novel about your formulation beyond applying it to the RFFI domain? Are there domain-specific characteristics of RF signals that make the inconsistency signal behave differently from prior OOD/OOSR studies?

---

### Note · Authors · 2026-01-05

I have read and agree with the venue's withdrawal policy on behalf of myself and my co-authors.